# Suicide and self-harm in low- and middle-income countries during the COVID-19 pandemic: A systematic review

**Duleeka Knipe**[1,2]*, **Ann John**[3], **Prianka Padmanathan**[1], **Emily Eyles**[1], **Dana Dekel**[3], **Julian P. T. Higgins**[1,4], **Jason Bantjes**[5,6], **Rakhi Dandona**[7,8], **Catherine Macleod-Hall**[1], **Luke A. McGuinness**[1], **Lena Schmidt**[1,9], **Roger T. Webb**[10,11], **David Gunnell**[1,4]

1 Population Health Sciences, Bristol Medical School, University of Bristol, Bristol, United Kingdom, 2 South Asian Clinical Toxicology Research Collaboration, Faculty of Medicine, University of Peradeniya, Peradeniya, Sri Lanka, 3 Population Data Science, Swansea University Medical School, Swansea, United Kingdom, 4 National Institute of Health Research Biomedical Research Centre, University Hospitals Bristol and Weston NHS Foundation Trust, Bristol, United Kingdom, 5 Alcohol, Tobacco and Other Drug Research Unit, South African Medical Research Council, Cape Town, South Africa, 6 Institute for Life Course Health Research, Department of Global Health, Stellenbosch University, Stellenbosch, South Africa, 7 Public Health Foundation of India, Gurugram, India, 8 Institute for Health Metrics and Evaluation, University of Washington, Seattle, Washington, United States of America, 9 Sciome LLC, Research Triangle Park, Durham, NC, United States of America, 10 Division of Psychology & Mental Health, University of Manchester, Manchester, United Kingdom, 11 National Institute of Health Research Greater Manchester Patient Safety Translational Research Centre, Manchester, United Kingdom

* Dee.knipe@bristol.ac.uk

**Data Availability Statement:** All data are available within the manuscript.

**Funding:** DK was supported by the Wellcome Trust through an Institutional Strategic Support Fund

## Abstract

There is widespread concern over the potential impact of the COVID-19 pandemic on suicide and self-harm globally, particularly in low- and middle-income countries (LMIC) where the burden of these behaviours is greatest. We synthesised the evidence from the published literature on the impact of the pandemic on suicide and self-harm in LMIC. This review is nested within a living systematic review (PROSPERO ID CRD42020183326) that continuously identifies published evidence (all languages) through a comprehensive automated search of multiple databases (PubMed; Scopus; medRxiv, PsyArXiv; SocArXiv; bioRxiv; the WHO COVID-19 database; and the COVID-19 Open Research Dataset by Semantic Scholar (up to 11/2020), including data from Microsoft Academic, Elsevier, arXiv and PubMed Central.) All articles identified by the 4th August 2021 were screened. Papers reporting on data from a LMIC and presenting evidence on the impact of the pandemic on suicide or self-harm were included. Methodological quality was assessed using an appropriate tool, and a narrative synthesis presented. A total of 22 studies from LMIC were identified representing data from 12 countries. There was an absence of data from Africa, the Pacific, and the Caribbean. The reviewed studies mostly report on the early months of COVID-19 and were generally methodologically poor. Few studies directly assessed the impact of the pandemic. The most robust evidence, from time-series studies, indicate either a reduction or no change in suicide and self-harm behaviour. As LMIC continue to experience repeated waves of the virus and increased associated mortality, against a backdrop of vaccine inaccessibility and limited welfare support, continued efforts are needed to track the indirect impact of the pandemic on suicide and self-harm in these countries.

Award to the University of Bristol [204813] and the Elizabeth Blackwell Institute for Health Research, University of Bristol. DG and JPTH are supported by the NIHR Biomedical Research Centre at University Hospitals Bristol and Weston NHS Foundation Trust and the University of Bristol. JPTH is supported by NIHR Applied Research Collaboration West at University Hospitals Bristol and Weston NHS Foundation Trust. JPTH is a National Institute for Health Research (NIHR) Senior Investigator (NF-SI-0617-10145). LAM is supported by an NIHR Doctoral Research Fellowship (DRF-2018–11-ST2-048). The funders had no role in study design, data collection and analysis, decision to publish, or preparation of the manuscript.

**Competing interests:** The authors have declared that no competing interests exist.

# Introduction

There is major concern over the impact of the COVID-19 pandemic and associated public health measures on mental health, in particular a potential rise in suicide and self-harm. Analysis of data from 21 higher income countries indicated that there was no evidence of an increase in suicide death rates during the early months of the pandemic [1]. The inclusion criteria for the reported multi-country synthesis required data from an official source spanning at least 16 months pre-pandemic. This meant that data from low- and middle-income countries (LMIC) were largely excluded as timely data from these were not readily available. Whilst this multi-country study indicated no increase in suicide deaths, these 21 countries do not represent regions that account for over 70% of global suicide deaths [2]. These settings have limited critical care services, mental health service provision, and welfare support [3]. Initial lockdowns were often also enforced with inadequate basic resources (e.g. uninterrupted food supply chains, income support), and often left migrant workers stranded from their families both within countries and across international borders [4].

As many LMIC are now experiencing additional waves of infection [5], together with major challenges in accessing and delivering vaccines [6], lockdown measures are being reinstated. Unlike many high income countries (HIC) where wage and job protection schemes have been implemented by governments, LMIC lack the necessary fiscal resources to provide such support at scale or for extended periods of time [7]. In this context the long-term economic and mental health impact of the pandemic is likely to be far worse than in HIC. Furthermore, many LMIC economies are reliant on international tourism and remittances sent home by migrant workers, both of which have been severely affected by sharp reductions in travel and persistent barriers to migration [4, 8].

Given the likely varying effect of the pandemic and associated lockdowns on populations in LMIC compared with those in HIC, the impact on suicide and self-harm risk may also be different. This systematic review aims to summarise the existing published evidence on the impact of the COVID-19 pandemic and its associated public health measures on suicide and self-harm in LMIC.

# Methods

## Protocol and registration

The current systematic review is nested within a larger registered living systematic review (PROSPERO ID CRD42020183326; registered on 1st May 2020) with similar inclusion and exclusion criteria also applied for this review [9].

## Eligibility criteria

The exposure of interest was the COVID-19 period and related experiences. The COVID-19 period was defined based on the authors' definition in the included papers. The related experiences include physical distancing, quarantine, lockdown, school and university closures, stigma, being infected by the virus, being in contact with someone with the virus, COVID-19 related bereavement and any other relevant exposure on suicidal behaviour/thoughts. We also include studies that report on factors that may have reduced the risk of self-harm or suicidal behaviour (e.g. increased belonging/social connectedness). The comparison is a pre-COVID-19 period, and/or individuals who have not been exposed to the related experiences outlined above. As such, cross-sectional studies that reported only on the prevalence of suicide and/or self-harm during the COVID-19 pandemic were excluded. The outcomes of interest were suicide deaths and self-harm (with or without suicidal intent). Most studies did not explicitly

differentiate between non-fatal suicidal behaviour (i.e., suicide attempts) and non-suicidal self-injury/harm. If a formal assessment of intent for non-fatal suicide attempts was not made these are presented as self-harm studies in this review. We present acts as described by study authors for all included studies. All study designs were included, and no exclusions were made based on language. Single case reports were excluded.

The only deviation from the living review protocol was that this nested review excluded studies based in HIC [10], those focused on suicidal thoughts only, and studies which exclusively rely on media reports of cases of fatal and non-fatal suicidal behaviour and/or self-harm (e.g. [11]). However, papers which reported on official data sources of suicide (e.g. police statistics) and/or self-harm in the media were included (e.g. [12]). Papers published and with complete expert review between 1st January 2020 and 4th August 2021 were included. If a single study or data source was reported in multiple published outputs, only the article with the most comprehensively reported data was included in the review. Where data from several countries were reported in a single paper, if data were extractable for LMIC separately these were included.

## Information sources

We searched PubMed, Scopus, medRxiv, bioRxiv, the COVID-19 Open Research Dataset by Sematic Scholar, and the Allen Institute for AI, which includes relevant records from Microsoft Academic, Elsevier, arXiv and PubMed Central (up to 11/2020); and the WHO COVID-19 database. Full details of searches are published elsewhere [9]. Both peer-reviewed and pre-print publications were included.

## Study selection

Within the broader living systematic review (https://covid19-suicide-lsr.info/), titles and abstracts were screened by a single reviewer and where there was uncertainty the full paper was viewed. One hundred randomly sampled outputs were subsequently rescreened (blind) by DG–there was complete agreement for all excluded studies. Full texts were assessed for eligibility by either DG, AJ, RTW, or DK for the living review. DK randomly screened studies assessed for eligibility by DG, AJ and RTW (10 studies for each reviewer), with high level agreement amongst the reviewers (Kappa = 0.87; 95% CI 0.69, 1.00). For this nested review, DK reassessed all included papers for the additional exclusion criteria (see above). All screening was done using a purpose-built online platform (Shiny web app, supported by a MongoDB database). All eligible articles that were not written in English were reviewed, with relevant data extracted with the help of someone fluent in that language and Google Translate [13].

## Data extraction

Using a piloted structured data extraction form, data were extracted initially by one review author (DG, AJ, or RTW), and independently repeated by DK. Data on study characteristics (country, setting, design, observations period(s), number of participants), details of lockdown or other societal restrictions, outcomes (suicide and/or self-harm), key findings, and strengths/limitations were extracted.

## Quality assessment

We used the Joanna Briggs Institute critical appraisal checklist that was relevant to each included study, except for time-series and before and after studies. For time-series studies we used the risk of bias criteria suggested by the Cochrane Effective Practice and Organisation of

Care [14]. For before and after studies we used an adapted version of the National Institute of Health quality assessment tool for this design [15]. We aimed to use these tools to assess overall study quality. Whilst each study was assessed for all criteria listed in the tools, we categorised studies as being of reasonable if they met specific criteria (see supplementary material). Quality assessments were carried out by two study authors (DK and PP) independently and a consensus rating (i.e., whether the study was of reasonable quality or not) was generated. All studies were included in the synthesis regardless of study quality. When a single study reported on multiple countries, each LMIC data source was assessed separately.

## Other reported data

To provide context for the review we have presented the estimated case counts and suicide rates in 2019 derived from the Global Burden of Diseases study for each country included in the review [2].

## Analysis

The impact of the pandemic is likely to have had different effects in each of the countries included in this review. The stringency and duration of lockdown measures implemented, varying availability of fiscal resources, and differences in pre-pandemic suicide rates/burden and legal status of suicidal behaviour between the countries included in this review indicate that a meta-analysis of all LMIC studies would be inappropriate. Due to multiple sources of heterogeneity, we have not conducted a meta-analysis but have presented a narrative synthesis of the data. We present the findings by World Bank regions and describe the findings firstly for suicide and then for self-harm. Where possible we also present unadjusted estimates of changes in the rate of suicide (fatal/non-fatal) and/or self-harm during the pandemic compared with a pre-pandemic period. For studies that reported estimates for a pre-pandemic period as well as the COVID-19 period, we have calculated a rate ratio with associated confidence intervals, using the *csi* command in STATA. As these studies did not provide an estimate of the size of the underlying population from which the cases of self-harm or suicide were sampled from, we assumed a stable population during the pre-pandemic and COVID-19 period. Under this assumption, the population term cancels out when calculating the risk ratio. For the studies where we calculated rate ratios, we provide the number of cases pre- and during the pandemic (supplementary material). We use this to provide a visual representation (i.e., a forest plot) of the data in comparison with other evidence included in this review.

## Results

Twenty-two studies met our eligibility criteria (Fig 1) and included data from 12 countries (9% of 135 LMIC–Fig 2). These countries contribute 68% of all suicide deaths in LMIC in 2019 (Table 1). The latest date for which suicide or self-harm outcomes were reported in any study was October 2020 [1]. Studies included are described below by World Bank region. There were studies representing all but one World Bank region, Africa (Table 2). Full quality assessment ratings are available as a supplement.

## East Asia & pacific

China (n = 6 studies) and Thailand (n = 1) were the only countries with eligible publications for the East Asia and Pacific region [16–22]; two studies from China [18, 21] were rated as being of reasonable quality. There were no studies specifically from the Pacific region.

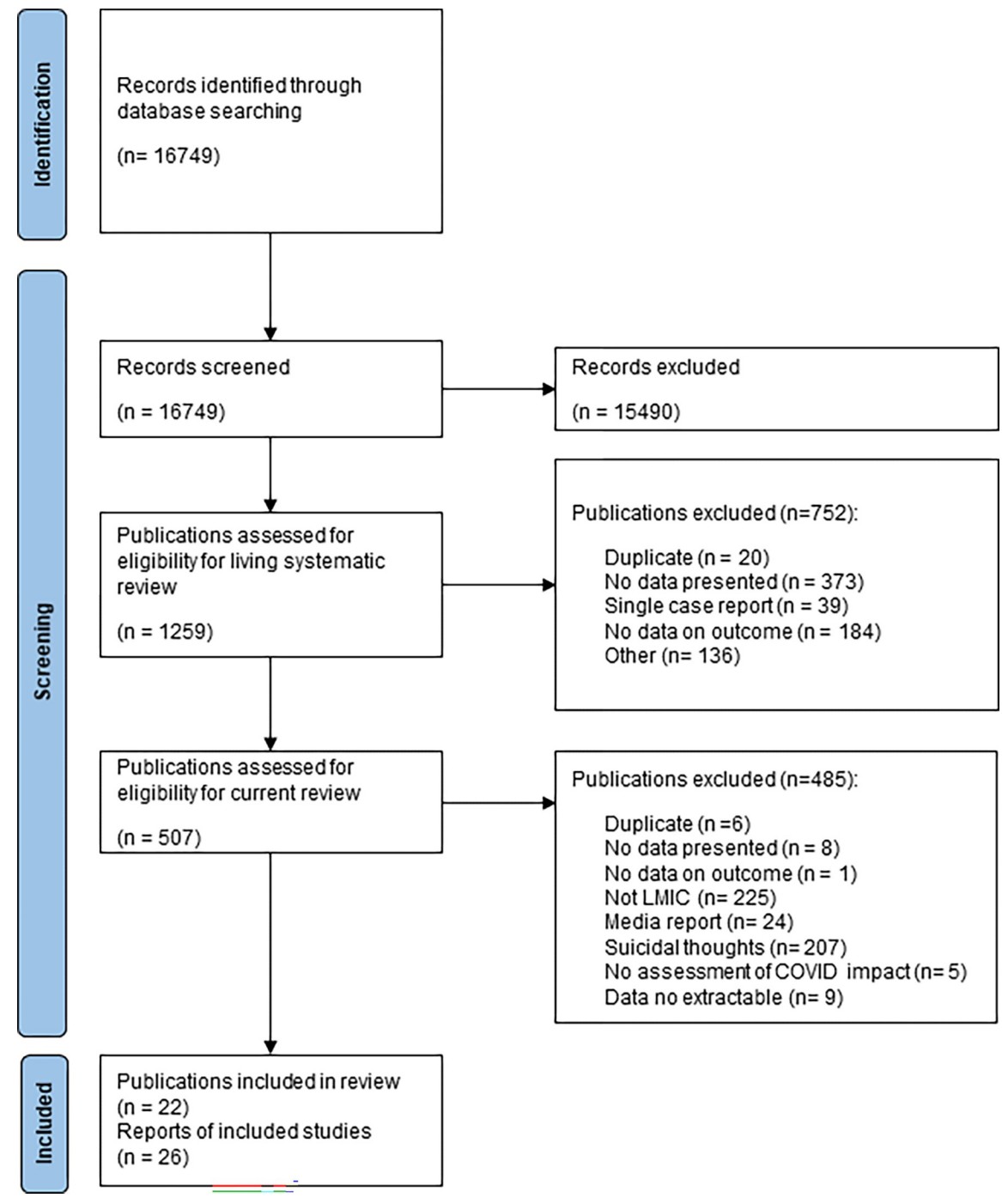

**Fig 1. PRISMA 2020 flow diagram of study selection.**

There were only two studies investigating the effect of the pandemic on suicide rates. One of these studies reported a decline in the suicide rate in Guangdong province, China from 3.73 per 100,000 before COVID-19 versus 3.04 per 100,000 (p<0.05) during the pandemic [21] (Fig 3). The decline was observed in both sexes, but age-stratified analyses suggested an increase in the suicide rate at ages 10–14 years. The second study, from Thailand, indicated that there was

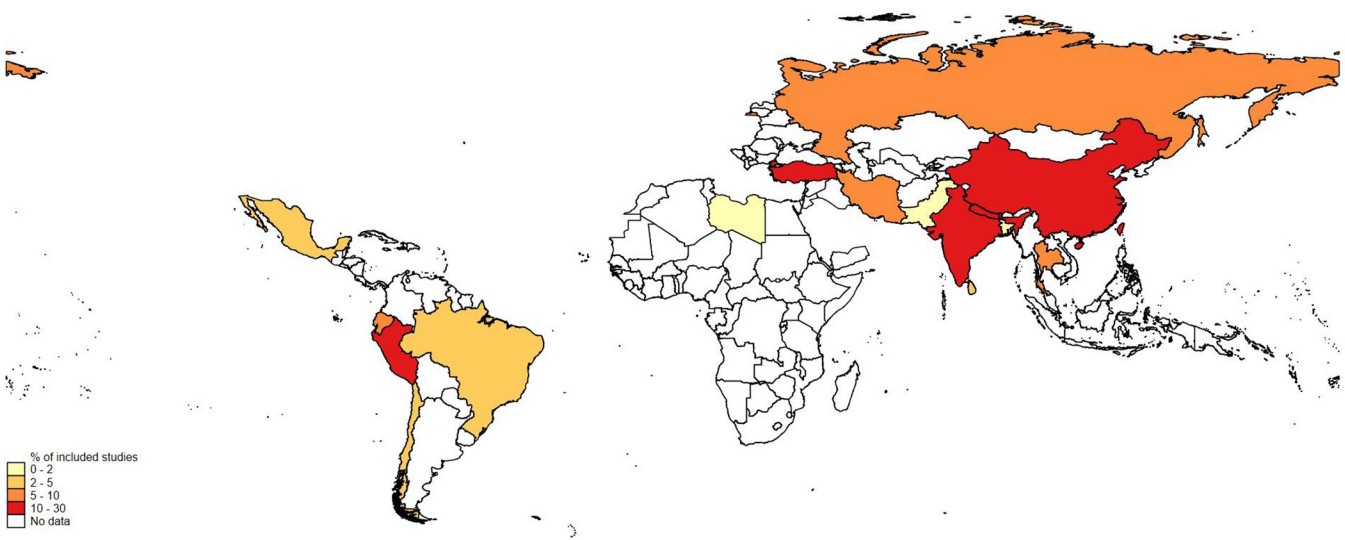

**Fig 2. Distribution of evidence from low- and middle-income countries on the impact of COVID-19 on suicidal behaviour.** The base map layer used to construct the map was obtained from Esri (https://hub.arcgis.com/datasets/esri::world-countries-generalized/about). High income countries are not represented in the map.

evidence of more than a 20% rise in suicide deaths between July 2019 and June 2020. This was a very brief report based on data from a secondary source (media report of police suicide statistics) and the quality and validity of the data is difficult to assess as we were unable to obtain the original data on which the report was based [22]. There was no statistical analysis conducted.

The remaining four studies from China investigated self-harm [16, 18–20], with one study explicitly reporting on non-suicidal self-injury (NSSI) [20]. This before and after study of 1241 Chinese primary school students reported a doubling in risk of self-harm (OR 2.20 95% CI

**Table 1. Number of suicide deaths and rate of suicide deaths in 2019 for countries included in review, the global burden of disease study 2019 [2].**

|  |  | Number of suicide deaths | Suicide death rate per 100,000 population | % of all suicide deaths in LMIC |
|---|---|---|---|---|
| **East Asia and Pacific** | | | | |
|  | China | 121217 | 8.5 | 20.5 |
|  | Thailand | 7107 | 10.1 | 1.2 |
| **Europe and Central Asia** | | | | |
|  | Russian Federation | 39040 | 26.6 | 6.6 |
|  | Turkey | 2585 | 3.2 | 0.4 |
| **Latin America and Caribbean** | | | | |
|  | Brazil | 13503 | 6.2 | 2.3 |
|  | Ecuador | 1678 | 9.5 | 0.3 |
|  | Mexico | 7805 | 6.3 | 1.3 |
|  | Peru | 1016 | 3.00 | 0.2 |
| **Middle East and North Africa** | | | | |
|  | Iran | 4172 | 5.00 | 0.7 |
| **South Asia** | | | | |
|  | India | 195336 | 14.2 | 33.0 |
|  | Nepal | 3528 | 11.6 | 0.6 |
|  | Sri Lanka | 4425 | 20.3 | 0.8 |

**Table 2. Summary of included studies.**

| World Bank Region | Country (region/city) | Author (year) | Design | Outcome (description from study) | Setting/ Population | Study period | Quality rating |
|---|---|---|---|---|---|---|---|
| East Asia and Pacific | China (Hubei province) | Mei (2021) [16] | Cohort | Self-harm (suicide attempts) | Hospitalised COVID-19 patients (exposed); Community individuals (unexposed) | 18 Jan—28 Jul 2020 | Low |
| East Asia and Pacific | China (National) | Tong (2021) [18] | Cross-sectional | Self-harm (suicide attempts) | Callers to crisis hotline. | 25 Jan—15 Jul 2020 | Reasonable |
| East Asia and Pacific | Thailand (Bangkok) | Thongchuam (2021) [17] | Before & after | Self-harm (intentional injury) | Admission to surgical unit of a tertiary hospital for corrosive ingestion | Jul 2019—Jun 2020 | Low |
| East Asia and Pacific | China (Guangdong province) | Zheng (2021) [21] | Before & after | Suicide deaths | Mortality data from Chinese Centre for Disease Control and Prevention | 01 Jan 2019–30 Jun 2019; 01 Jan 2020–30 Jun 2020 | Reasonable |
| East Asia and Pacific | Thailand (National) | Ketphan (2020) [22] | Before & after | Suicide deaths | Police records from media reports** | 2019–2020 | Low |
| East Asia and Pacific | China (Anhui province) | Zhang (2020) [20] | Before & after | Self-harm (suicide attempts) and NSSI | Primary school students | 1st survey Nov 2019; 2nd survey May 2020 | Low |
| East Asia and Pacific | China (Wuhan) | Xu (2021) [19] | Cross-sectional | Suicide thoughts and self-harm* (suicide attempts) | Online survey of university students | 29 Jun—18 Jul 2020 | Low |
| Europe and Central Asia | Turkey (Ankara) | Fidanci (2021) [24] | Before & after | Self-harm ("suicide consultations") | Paediatric emergency department admissions | Apr 2019—Oct 2019; Apr 2020—Oct 2020 | Low |
| Europe and Central Asia | Turkey (Bursa) | Eray (2021) [23] | Before & after | Self-harm (admission reasons recorded as "hurting yourself" and "suicide attempt") | Admission to child and adolescent emergency department | 11 Mar—30 Sept 2019; 11 Mar—30 Sept 2020 | Low |
| Europe and Central Asia | Turkey (National) | Teksin (2020) [26] | Cross-sectional | Suicide thoughtsand self-harm* (suicide attempt) | Online survey of health care workers | 20 May—10 Jun 2020 | Low |
| Europe and Central Asia | Russian Federation (National) | Pirkis (2021) [1] | Time series | Suicide deaths | Prelim data from forensic medical examination | Jan 2016—Jul 2020 | Reasonable |
| Europe and Central Asia | Russian Federation (Moscow) | Gerasimova (2020) [25] | Before & after | Suicide and Self-harm (suicide attempts) | Calls to an emergency helpline | 01 Mar—30 Apr 2019; 01 Mar– 17 Apr 2020 | Low |
| Latin America and Caribbean | Brazil (Botucatu, Maceio) | Pirkis (2021) [1] | Time series | Suicide deaths | Death certificates that are completed by a medical doctor | Jan 2019—Sept 2020 | Reasonable |
| Latin America and Caribbean | Ecuador | Pirkis (2021) [1] | Time series | Suicide deaths | Police reports | Jan 2017—Oct 2020 | Low |
| Latin America and Caribbean | Mexico (Mexico City) | Pirkis (2021) [1] | Time series | Suicide deaths | Criminal record of suicide death from the Attorney General's Office | Jan 2019—Oct 2020 | Low |
| Latin America and Caribbean | Peru (National) | Pirkis (2021) [1] | Time series | Suicide deaths | Peruvian National Death Information System | Jan 2017—Sept 2020 | Reasonable |
| Middle East and North Africa | Iran (Tehran) | Forouzanfar (2021) [29] | Case series | Suicide deaths | Hospital emergency department | - | Low |
| Middle East and North Africa | Iran (Tehran) | Pirnia (2020) [30] | Case series | Suicide deaths | Unclear | Apr-20 | Low |

(*Continued*)

**Table 2.** (Continued)

| World Bank Region | Country (region/ city) | Author (year) | Design | Outcome (description from study) | Setting/ Population | Study period | Quality rating |
|---|---|---|---|---|---|---|---|
| South Asia | India (Burla) | Acharya (2020) [31] | Before & after | Self-harm (suicidal injuries) | Ear nose and throat hospital department | 1 Sept 2019–31 Aug 2020 | Low |
| South Asia | India (Chandigarh) | Sahoo (2020) [35] | Case series | Self-harm (self-harm) | Hospital emergency department | - | Low |
| South Asia | India (Rishikesh) | Jhanwar (2020) [33] | Before & after | Self-harm (suicide attempt) | Patients who had a hospital presenting psychiatric emergency | 24 Feb—23 Apr 2020 | Low |
| South Asia | Nepal (Kathmandu) | Shrestha (2021) [37] | Before & after | Self-harm (Self-harm/suicide attempts) | Hospital emergency department | 24 Mar—23 Jun 2019; 24 Dec 2019–23 Mar 2020; 24 Mar—23 Jun 2020 | Low |
| South Asia | Sri Lanka (Peradeniya) | Knipe (2021) [34] | Time series | Self-harm (self-poisoning) | Hospital admission to toxicology ward | 1 Jan 2019–31 Aug 2020 | Reasonable |
| South Asia | India (Cooch Behar) | Sengupta (2020) [36] | Before & after | Suicide deaths | Hospital autopsies | 25 Mar—24 Apr 2019; 25 Jan—24 Apr 2020 | Low |
| South Asia | India (New Delhi) | Behara (2021) [32] | Before & after | Suicide deaths | Hospital autopsies | 25 Mar—31 Oct 2019; 25 Mar—31 Oct 2020 | Low |
| South Asia | Nepal (National) | Poudel (2020) [12] | Before & after | Suicide deaths | Police records from media reports** | 23 Mar—06 Jun 2020 | Low |

* Aggregate measure

** Original data could not be accessed but authors have confirmed that these data do exist through communication with key contacts in both Nepal and Thailand.

1.56, 3.10) in the pandemic compared with a pre-pandemic period 6 months earlier [20]. This study also reported an elevated risk of NSSI (OR 1.55 95% CI 1.40, 1.72) during the pandemic period; however, the self-reported reference period for this measure during the pandemic included a pre-pandemic period (the 12 months before May 2020, whereas the onset of the pandemic in China was during January 2020). In addition, the statistical analysis of this study did not account for repeated outcome measurements from the same individual.

One study reported on the volume of crisis hotline calls during the early months of the pandemic (Jan–Jul 2020), with callers asked whether they had self-harmed in the 2 weeks prior to the call [18]. The calls were then categorised into COVID-19 related calls versus non-COVID-19 calls. This classification was made, by the operator, based on whether the caller indicated that they had been psychologically impacted by the pandemic, or encountered a problem which could be directly (e.g., infection) or indirectly (e.g. job loss) attributed to the pandemic. The study reported that non-COVID-19 callers were more likely than COVID-19 related callers to report self-harm in the preceding 2 weeks (p = 0.005). One key limitation of this analysis is that many callers who were categorised as being non-COVID-19 will have been impacted by the pandemic in ways that were not measured.

Two studies reported on suicide and self-harm risk in individuals infected with COVID-19 [16, 19]. The first was a cohort study that reported 4 cases of self-harm in 4328 COVID-19 patients in Hubei province, China over an average follow-up period of 144 days (median) compared to no reported self-harm in an unexposed community sample [16]. It is, however,

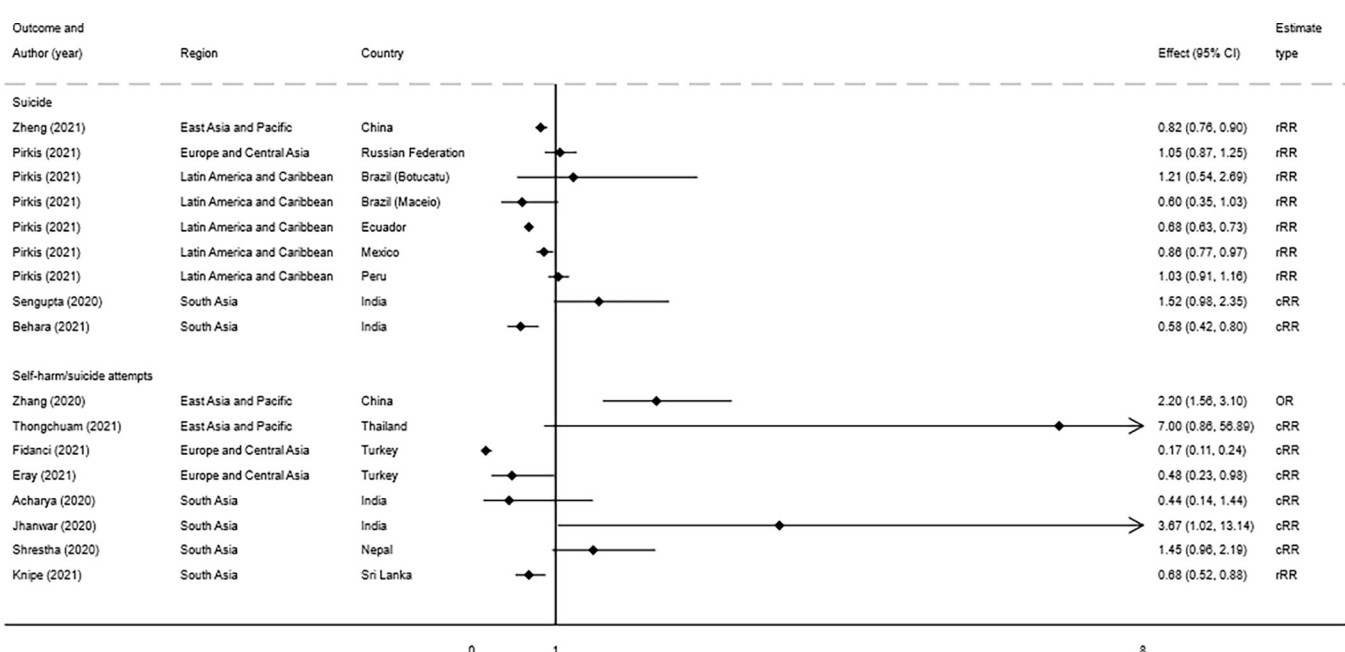

**Fig 3. Forest plot of study estimates*** **assessing the impact of the pandemic on suicide deaths and self-harm from before and after or time-series studies.** *
An estimate of below 1 is indicative of a reduction in suicide or self harm. rRR–reported rate ratio; cRR–calculate rate ratio; OR–Odds ratio.

unclear how self-harm was assessed and the likelihood of undetected COVID-19 infection in the community sample was also not considered. The second study was an online survey on university students that reported a self-harm prevalence of 0.1% (n = 11), which also found that individuals with confirmed or suspected COVID-19, or who were in close contact with a confirmed case, had a raised prevalence of suicidal thoughts and self-harm (as a combined outcome) [19]. Additionally, the study reported that changes in lifestyle, alcohol use, and high levels of stress during the pandemic were associated with elevated risk of suicidal thoughts and self-harm.

## Europe & Central Asia

Evidence from the Europe and Central Asia region included data from Turkey (n = 3 studies), and the Russian Federation (n = 2) [1, 23–26]. There was only 1 reasonable quality study, and this was the only study which directly reported on the impact of COVID-19 on suicide risk [1]. This study found no evidence that the pandemic had an impact on suicide death rates in the Russian Federation (Fig 3).

Another study from Russian Federation reported on suicide- and self-harm related calls to an emergency helpline during the pandemic compared to a pre-pandemic period [25]. The number of calls related to these outcomes were very small (n = 27: pre-pandemic = 4; pandemic = 23) and limited information was given as to how calls were coded.

Two studies reported on hospital emergency department visits by children and adolescents. The first study conducted in the Turkish city of Bursa found no evidence that the proportion of hospital admissions following emergency department presentation after self-harm, differed in the pandemic period compared with the same period in 2019. However, the absolute number of presentations dropped from 23 to 11 [23]. This was a similar finding to the second study conducted in Turkey's capital city, Ankara, which found no evidence that the proportion of

admissions for self-harm changed during the pandemic, although there were 83% fewer presentations (Fig 3) [24].

A cross-sectional study of Turkish healthcare workers found some evidence that those who reported suicidal thoughts and self-harm during the pandemic had a higher perceived stigma score than unaffected workers [26]. The stigma score was generated using an unvalidated questionnaire, and it is unclear what questions were asked. The author-generated questionnaire included questions to "identify the events experienced by health-care professionals during the pandemic and the feelings and thoughts they have experienced". Interpreting these findings is therefore challenging, but it is possible that perceived stigma of caring for COVID-19 patients might have induced suicidal thoughts and self-harm behaviour.

## Latin America & the Caribbean

There were four investigations from the Latin America and the Caribbean region, which included data only from the Latin American countries of Brazil, Ecuador, Mexico, and Peru from a single paper [1]. None of the investigations were from the Caribbean. Given limitations in the data sources used, only 2 of the investigations were rated as being of reasonable quality.

There were three Peruvian studies which utilised national register data on suicide deaths covering the years 2017 to 2020 [1, 27, 28]. We present the finding from the study with the highest quality evidence of risk spanning the longest time period [1]. This study reported no evidence of either an increase or decrease in the rate of suicide in the first 6 months of the pandemic (Fig 3) [1].

In the two countries, Ecuador and Mexico, with available data up until the end of October 2020, there was evidence of a decrease in the number of suicide deaths in the pandemic period (Fig 3) [1]. In Brazil there was no evidence of either an increase or decrease.

There were no papers reporting on the impact of the pandemic on non-fatal suicidal behaviour or self-harm from this region.

## Middle East & North Africa

Two case series of suicide deaths in Tehran (Iran) represent the only studies from this region of the world [29, 30], and therefore evidence from North Africa is lacking. Neither study was rated as being of reasonable quality. Both studies were based on two suicide deaths each (mother-son pairs) and reported that COVID-19 related bereavement was associated with the suicide deaths.

## Africa

There were no studies from LMIC in Africa.

## South Asia

India (n = 5), Sri Lanka (n = 1) and Nepal (n = 2) represented the South Asian region with 8 studies [12, 31–37], with only one (from Sri Lanka) rated as being of reasonable quality.

Several publications from Nepal have summarised media reporting of official (police data) suicide death statistics [12, 22, 38–40], and we hereby present the evidence from the most comprehensive publication [12]. The original data source was not accessible, but the report indicates that there were 20% more suicide deaths during the first month of the pandemic compared to a pre-pandemic period. The rate of suicide deaths per day during the pandemic (16.5 suicide deaths/day) was slightly higher than in 2019 (15.8 suicide deaths/day). Underlying trends of suicide were not considered.

A hospital autopsy study from two districts of New Delhi, India reported a reduction in suicide deaths in the lockdown period (March–May 2020; Fig 3), but no difference in the number of deaths in the post lockdown pandemic period (June–October 2020) [32]. A second hospital suicide autopsy study from the city of Cooch Behar, India reported that a higher proportion of autopsies conducted in the first month of lockdown (April 2020) were due to suicide compared with the same month in 2019 [36] (Fig 3). In neither of these studies from India were underlying trends taken into account.

Only one study from Nepal reported on changes in self-harm frequency during the pandemic period [37]. This study compared hospital emergency department presentations for self-harm during the first three months of the lockdown period (March-June) with the same time period in 2019. These data suggest an increase in the number of presentations for self-harm during the pandemic period compared to the same three-month period in 2019 (Fig 3), although antecedent trends were not considered. This study also reports evidence of an increased delay between self-harm and subsequent hospital presentation, and an elevated case fatality during the pandemic period.

Three hospital studies in India reported on self-harm, one case series and two before and after studies. The hospital case series indicated self-harm behaviour was a direct result of anxiety/fear resulting from COVID-19 related media reports in the two presented individuals [35]. One small study (n = 14) in Rishikesh, India reported an increase in suicide-related psychiatric emergencies in the 4 weeks after lockdown compared with the preceding 4 weeks (11 vs 3—Fig 3) [33]. Another small hospital study (n = 13) from the district of Burla, in the Indian city of Sambalpur, reported the number of suicidal cut-throat injuries increased from 4 in the 6 months prior to the pandemic to 9 in the following 6 months (Fig 3) [31].

There was a single study from Sri Lanka that reported a 32% reduction in hospital presentations for self-poisoning during the pandemic period compared to pre-pandemic trends (RR 0.68 95% CI 0.52, 0.88) (Fig 3) [34]. There was no evidence that the apparent impact of the pandemic differed by sex or age.

## Discussion

In this systematic review, we found only 22 studies reporting on suicide deaths and self-harm during the COVID-19 pandemic from only 9% (n = 12) of all LMIC (n = 135), with a complete absence of evidence from African countries. The evidence-base was mostly methodologically poor (77%), with studies generally lacking comparator data to enable an assessment of whether the observed rates or differences are specifically related to the pandemic. The exceptions to this were a repeated cross-sectional study from China and time series analyses from Brazil, China, Ecuador, Mexico, Peru, Russian Federation, and Sri Lanka. The Chinese repeated cross-sectional study found an increased odds of self-harm and NSSI during the pandemic period in primary school pupils compared with a few months before the pandemic. However, this finding should be interpreted with caution, as the incidence of suicidal behaviours varies seasonally and increases rapidly at this age (e.g. puberty effects) [41], and the rise observed may reflect the ageing of the sample rather than a causal pandemic effect. The time series analyses provide the most robust evidence included in this review, and these studies consistently show either no evidence of an impact or a decrease in suicide deaths and self-harm during the pandemic period. This is similar to the effect observed in HIC [1]. There was some indication from China that the impact may differ by age [21]; whilst an overall reduction was observed there was evidence that suicide rates increased in young people.

The disparity in research evidence against the burden of suicide deaths globally has been previously documented [42]–less than 20% of the studies identified as part of the wider living

systematic review [9] pertain to LMIC. Despite India accounting for the largest number of suicide deaths prior to the pandemic, there were no reasonable quality studies from this country. In addition, even though the Russian Federation has one of the highest suicide rates globally (top 5%) [2], there was only one (low quality) study from this country. The absence of data from the Africa region is not surprising as suicide and self-harm prevention research has historically not been a priority in the region. Typically, countries in Africa experience serious healthcare resource constraints and face several chronic pressing health challenges, including a high burden of infectious diseases and maternal mortality. Suicide prevention is not considered the most important public health problem in the region, and consequently does not receive the same attention as in HIC. Any impact that the COVID pandemic has had on suicide and self-harm rates in Africa is likely to be largely obscured by more widespread visible health problems and the increases in mortality from other diseases whose management has been affected by the pandemic, and the shutting down of the already limited mental healthcare services in order to re-deploy medical staff to emergency and intensive care units for COVID patients.

Even though it is well known that rates of suicide and self-harm vary by gender [43, 44], and that the indirect impact of the pandemic is likely to differ by gender, this gender perspective was largely overlooked in the studies included. Without this perspective our understanding of the effect of the pandemic on suicide and self-harm is hindered, and hampers planning of appropriate action.

Many of the studies (> 80%) included in this synthesis only assessed the impact during the early stages of the pandemic (i.e. the first 5-months of the crisis)–the most current data included in this review were from October 2020, despite our final search date being the 4[th] August 2021. The limited findings from these studies are likely to be outdated by the relatively recent rises in cases and deaths of COVID-19 [5]. Since the searches were completed for the current review, a national suicide mortality data for all of India in 2020 have been published [45]; there was a 10% rise in recorded suicide deaths in 2020 compared to 2019, and this pre-dates the large surge in COVID-19 cases in India during April-June 2021. This further highlights the need for greater use of pre-print services and open science practices to ensure the timely dissemination of research during a public health crisis. In addition, unlike HIC where vaccine rollouts have been relatively successful, thus allowing for an easing of lockdown measures and a return to relative normality, LMIC have experienced difficulties in accessing vaccines and their populations are now experiencing new periods of lockdowns. As previously, these repeat lockdowns are being largely enforced with limited governmental fiscal support of business or welfare support, although there are exceptions to this (e.g., cash transfer programme in India). There is a need for continued effort in tracking the long-term impact of the pandemic on suicide and self-harm in LMIC where the burden of these behaviours is greatest. The need for reliable real-time surveillance systems in LMIC has always been important [46, 47], but is ever more pressing now—researchers, practitioners and policymakers need to renew their efforts to establish adequately resourced surveillance.

This systematic review has several strengths. It provides a contemporary synthesis of the evidence base identified from searching a wide range of databases without any language restrictions and includes assessments of study quality that are appropriate to each study design. Whilst we searched multiple databases, a limitation is that these tended to index primarily English language journals and therefore we may have missed important publications from regions in which English is not the first language. Furthermore, any literature review such as this is prone to publication bias if the studies that reach publication are not representative of those that have been conducted.

In the absence of reliable epidemiological data on suicide and self-harm in LMIC, it is impossible to assess the impact of the COVID-19 pandemic on these behaviours and to plan evidence-based prevention approaches. The lack of infrastructure and data on these behaviours has been longstanding, but the pandemic has brought this underinvestment into sharp focus. International organisations (e.g. World Health Organisation) need to support and invest in data collection in these countries.

Whilst there is a need for more research and evidence to help track suicide and self-harm and support prevention, governments and policy makers also need to be proactive in their prevention efforts acting on the existing evidence base [48]. Government and health systems should focus on: i) provision of economic supports and active labour market schemes, which may require foreign assistance; ii) working with the media to report responsibly; iii) ensuring protection measures are in place for victims of domestic violence; and iv) improving access to services and support charities for individuals who are suicidal or who are at elevated risk of becoming suicidal. Early signals from HIC indicate that the mental health of children and young people may have been particularly affected and their education interrupted [49], this may equally be the case in LMIC and merits special attention.

## Supporting information

**S1 Checklist. PRISMA reporting checklist for abstract.**
(DOCX)

**S2 Checklist. PRISMA reporting checklist for main manuscript.**
(DOCX)

**S1 Text. Further details of searches conducted, quality assessment, and the number of suicide and self-harm cases used to calculate rate ratios.**
(DOCX)

**S1 Data. All underlying data.**
(XLSX)

## Acknowledgments

The authors would like to thank Aliya Sarmanova for helping with data extraction and Chris Metcalfe for statistical advice. Chukwudi Okolie, Babatunde K. Olorisade and Hung-Yuan Cheng for help with other aspects of the Living Review.

## Author Contributions

**Conceptualization:** Duleeka Knipe, Ann John, Julian P. T. Higgins, David Gunnell.

**Data curation:** Duleeka Knipe, Ann John, Emily Eyles, Dana Dekel, Catherine Macleod-Hall, Roger T. Webb, David Gunnell.

**Formal analysis:** Duleeka Knipe, Julian P. T. Higgins, David Gunnell.

**Investigation:** Duleeka Knipe, Ann John, Emily Eyles, Dana Dekel, Catherine Macleod-Hall, Luke A. McGuinness, Lena Schmidt, Roger T. Webb, David Gunnell.

**Methodology:** Duleeka Knipe, Ann John, Julian P. T. Higgins, Luke A. McGuinness, Lena Schmidt, Roger T. Webb, David Gunnell.

**Project administration:** Duleeka Knipe, Ann John, Lena Schmidt, David Gunnell.

**Software:** Luke A. McGuinness.

**Supervision:** Lena Schmidt.

**Validation:** Ann John, Prianka Padmanathan.

**Visualization:** Duleeka Knipe.

**Writing – original draft:** Duleeka Knipe.

**Writing – review & editing:** Duleeka Knipe, Ann John, Prianka Padmanathan, Emily Eyles, Dana Dekel, Julian P. T. Higgins, Jason Bantjes, Rakhi Dandona, Catherine Macleod-Hall, Luke A. McGuinness, Lena Schmidt, Roger T. Webb, David Gunnell.

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
