## [Decision Letter · Decision Letter 0]

23 Dec 2021

PGPH-D-21-00865

Suicide and self-harm in low- and middle- income countries during the COVID-19 pandemic: A systematic review

Dear Dr. Kniipe,

Thank you for submitting your manuscript to PLOS Global Public Health. After careful consideration, we feel that it has merit but does not fully meet PLOS Global Public Health’s publication criteria as it currently stands. Therefore, we invite you to submit a revised version of the manuscript that addresses the points raised during the review process.

We look forward to receiving your revised manuscript.

Kind regards,

Ahmed Waqas

Academic Editor

Journal Requirements:

1. We ask that a manuscript source file is provided at Revision. Please upload your manuscript file as a .doc, .docx, .rtf or .tex. If you are providing a .tex file, please upload it under the item type ‘LaTeX Source File’ and leave your .pdf version as the item type ‘Manuscript’

Reviewers' comments:

Reviewer's Responses to Questions

**Comments to the Author**

1. Does this manuscript meet PLOS Global Public Health’s publication criteria? Is the manuscript technically sound, and do the data support the conclusions? The manuscript must describe methodologically and ethically rigorous research with conclusions that are appropriately drawn based on the data presented.

Reviewer #1: Yes

Reviewer #2: Yes

2. Has the statistical analysis been performed appropriately and rigorously?

Reviewer #1: Yes

Reviewer #2: Yes

3. Have the authors made all data underlying the findings in their manuscript fully available (please refer to the Data Availability Statement at the start of the manuscript PDF file)?

Reviewer #1: Yes

Reviewer #2: Yes

4. Is the manuscript presented in an intelligible fashion and written in standard English?

Reviewer #1: Yes

Reviewer #2: Yes

5. Review Comments to the Author

Reviewer #1: Where Google translate was used, it would be helpful to list this as a methodological limitation. Google Translate, while an incredibly beneficial tool, does not represent a fully faithful translation of text, especially regarding subjects as nuanced as suicidal/self-harm behaviour. This may be stylistic, but the authors' choice of WB regions conflates East Asia and the Pacific, and the dearth of eligible publications from the Pacific, which has a high rate of suicide and self-harm, makes their absence particularly striking. I don't think this is a methodological weakness, but it is a limitation in scope and should be presented as such.

Re: Thailand study: the authors have chosen to exclude from their analysis media reports of suicide/self-harm (and wisely, I might note). However, the study they cite was a media report, albeit of police suicide statistics. The authors should note in the article or in a footnote whether or not they attempted to locate the primary source and why this publication's inclusion is different from other excluded media reports. This is a methodological weakness as it is.

Reviewer #2: This systematic review is an important contribution to literature, and will be of interest to the policy makers. The systematic review adheres to standard guidelines and reports a robust methodology. All of the data associated with the review has been provided, for transparency and reproducibility.

I do not have any comment to add at this point.

6. PLOS authors have the option to publish the peer review history of their article (what does this mean?). If published, this will include your full peer review and any attached files.

**Do you want your identity to be public for this peer review?** For information about this choice, including consent withdrawal, please see our Privacy Policy.

Reviewer #1: No

Reviewer #2: **Yes: **Hafsa Meraj

---

## [Decision Letter · Decision Letter 1]

5 Apr 2022

Suicide and self-harm in low- and middle- income countries during the COVID-19 pandemic: A systematic review

PGPH-D-21-00865R1

Dear Kniipe,

We are pleased to inform you that your manuscript 'Suicide and self-harm in low- and middle- income countries during the COVID-19 pandemic: A systematic review' has been provisionally accepted for publication in PLOS Global Public Health.

Best regards,

Ahmed Waqas

Academic Editor

Reviewer Comments (if any, and for reference):

Reviewer's Responses to Questions

**Comments to the Author**

1. If the authors have adequately addressed your comments raised in a previous round of review and you feel that this manuscript is now acceptable for publication, you may indicate that here to bypass the “Comments to the Author” section, enter your conflict of interest statement in the “Confidential to Editor” section, and submit your "Accept" recommendation.

Reviewer #1: All comments have been addressed

Reviewer #2: All comments have been addressed

2. Does this manuscript meet PLOS Global Public Health’s publication criteria? Is the manuscript technically sound, and do the data support the conclusions? The manuscript must describe methodologically and ethically rigorous research with conclusions that are appropriately drawn based on the data presented.

Reviewer #1: Yes

Reviewer #2: Yes

3. Has the statistical analysis been performed appropriately and rigorously?

Reviewer #1: Yes

Reviewer #2: Yes

4. Have the authors made all data underlying the findings in their manuscript fully available (please refer to the Data Availability Statement at the start of the manuscript PDF file)?

Reviewer #1: Yes

Reviewer #2: Yes

5. Is the manuscript presented in an intelligible fashion and written in standard English?

Reviewer #1: Yes

Reviewer #2: Yes

6. Review Comments to the Author

Reviewer #1: Methodological concerns raised in my first review have been addressed. This study is timely and brings crucial attention to the need for more and more rigorous studies into the incidence of self harm and suicidality, especially in Africa.

Reviewer #2: All of my comments were addressed in previous iteration. I am happy to recommend acceptance of the manuscript.

7. PLOS authors have the option to publish the peer review history of their article (what does this mean?). If published, this will include your full peer review and any attached files.

**Do you want your identity to be public for this peer review?** For information about this choice, including consent withdrawal, please see our Privacy Policy.

Reviewer #1: **Yes: **Alexander Plum

Reviewer #2: **Yes: **Dr. Hafsa Meraj
